# Research on Lightweight Microservice Composition Technology in Cloud-Edge Device Scenarios

**DOI:** 10.3390/s23135939

**Published:** 2023-06-26

**Authors:** Hanqi Li, Xianhui Liu, Weidong Zhao

**Affiliations:** School of Electronics and Information Engineering, Tongji University, Shanghai 200092, China; lihanqi0717@icloud.com (H.L.); wd@tongji.edu.cn (W.Z.)

**Keywords:** workflow, microservice, workflow engine, industrial internet, edge computing, cloud edge combination

## Abstract

In recent years, cloud-native technology has become popular among Internet companies. Microservice architecture solves the complexity problem for multiple service methods by decomposing a single application so that each service can be independently developed, independently deployed, and independently expanded. At the same time, domestic industrial Internet construction is still in its infancy, and small and medium-sized enterprises still face many problems in the process of digital transformation, such as difficult resource integration, complex control equipment workflow, slow development and deployment process, and shortage of operation and maintenance personnel. The existing traditional workflow architecture is mainly aimed at the cloud scenario, which consumes a lot of resources and cannot be used in resource-limited scenarios at the edge. Moreover, traditional workflow is not efficient enough to transfer data and often needs to rely on various storage mechanisms. In this article, a lightweight and efficient workflow architecture is proposed to optimize the defects of these traditional workflows by combining cloud-edge scene. By orchestrating a lightweight workflow engine with a Kubernetes Operator, the architecture can significantly reduce workflow execution time and unify data flow between cloud microservices and edge devices.

## 1. Introduction

With the rapid development of technologies, such as cloud-native [1] and the Internet of Things, users have increasingly diverse demands for software systems. A service-oriented architecture [2] needs to strike a balance between stable service integration and flexible adaptation to user requirements. Therefore, microservices [3,4], which can run as independent processes and be deployed independently [5], have emerged as a promising solution.

At the same time, industrial microservices has been the carriers of IIoT (Industrial Internet of Things) platforms, which are software architectures based on modular combinations of single-function components to achieve “loose coupling” application development. A microservice is a small, independently deployable application that is designed to perform a single function. By combining multiple isolated microservices with different functions as needed [6] and enabling them to communicate with each other through APIs [7], a complete large-scale application system [8,9] can be constructed.

Currently, the construction of the IIoT in China is still in a stage of rapid development. However, small and medium-sized enterprises face certain challenges in the process of digital transformation and accessing the industrial Internet.

Firstly, when facing small and medium-sized enterprises in different vertical industries, few IIoT platforms can provide comprehensive solutions tailored to specific industries. Secondly, current IIoT applications are generally based on microservice architectures, which have complex installation processes and are difficult to apply quickly in small and medium-sized enterprises. Thirdly, the batch deployment and maintenance of microservice applications are challenging and require a large number of personnel with relevant microservice expertise, which may be unaffordable for small and medium-sized enterprises. Fourthly, edge devices come in various forms and have complex protocols, making it difficult to combine them with cloud applications. In conclusion, microservice composition is the core problem that needs to be solved for the rapid implementation of IIoT applications, and there is not yet enough in-depth research in this area.

In this context, this article proposes a cloud-edge combined lightweight microservice composition solution aimed at addressing process control [10]. The solution aims to integrate independent industrial microservice devices, including hardware devices and intelligent sensor terminals, with cloud microservice applications using cloud-native operating system extension technology [11]. The solution manages a self-developed lightweight microservice composition engine [12] and builds a cloud-edge combined microservice composition platform to address the pain points of slow development, complicated deployment, and complex maintenance in small and medium-sized enterprises’ digital transformation in the IIoT. The design of this article fully considers the limited conditions of edge resources, treats workflow as a resource, reduces workflow nodes from containers to user-level threads, and efficiently connects cloud services and edge devices in cloud-edge scenarios.

## 2. Related Work

### 2.1. Cloud-Edge Integration

Cloud-edge integration refers to the ability to monitor and manage distributed cloud, edge, and endpoint resources on a unified platform based on resource integration. It enables a unified perspective on operation and maintenance capabilities, simplifies user operations to the greatest extent possible, and enables intelligent inspection, alarm, upgrade, governance, and other capabilities for terminal devices, effectively reducing costs.

Research by Qiang Du and others [13] explores the concept, architecture framework, related technologies, and future development directions of cloud-edge integration. In cloud-edge integration, achieving efficient resource management and coordination, as well as innovative and optimized business models, are essential challenges.

### 2.2. Business Composition Modes

In cloud-native and distributed scenarios, more applications are split into multiple independent microservices that can be independently deployed, scaled, and updated [14,15]. However, dispersed microservices can also bring problems in terms of protocols, communication, and orchestration [16,17,18]. In this context, exploring a concise and efficient business composition mode has become a vital issue in the application orchestration field.

In the joint announcement of the Open Application Model (OAM) by Alibaba Cloud and Microsoft, a design concept of separation of concerns is proposed. The goal is for platform architects to quickly encapsulate the platform’s operation and maintenance capabilities into reusable components, allowing the application’s developers to focus on integrating these operation and maintenance components with their code. Based on this goal, the OAM standard proposes concepts such as application components, application deployment configuration files, and features for application operation and maintenance to define a platform-independent, highly scalable application description capability.

Based on the design concept of separation of concerns, we constructed a set of application deployment and release composition systems, attempting to shield the complexities of the underlying infrastructure operating environment and provide a concise and efficient microservice composition application management experience.

### 2.3. Workflow Composition Model

Currently, mainstream cloud-native workflows are based on native CICD (Continuous Integration and Continuous Delivery) workflows [19]. Cloud-native CICD workflows are used for building and deploying applications in a cloud-native architecture. To ensure that different applications do not interfere with each other, each workflow node often spins up a separate container. This design has many advantages in a continuous integration environment, but when it comes to the combination of applications and processes, spinning up and recycling containers can consume a lot of machine resources, leading to slow workflow execution. Therefore, this CICD workflow composition model has severe performance defects in application composition execution.

### 2.4. Optimization for Cloud Edge Scenarios

Research by Prohim Tam and others [20] mainly focuses on optimizing the offloading process of multi-service tasks in federated learning. The authors propose an intelligent agent based on deep reinforcement learning, which combines software-defined networking and network function virtualization to offload tasks to edge computing nodes in order to improve the efficiency of federated learning in edge environments. The article introduces the system model, including communication and computation aspects, and proposes an optimized multi-service task offloading method. By considering factors such as completion time, energy consumption, and round communications, the article designs a reward mechanism to optimize the construction of the global model. Experimental results show that the method achieves significant effects in reducing communication overhead and improving system performance.

Research by Caihong Kai and others [21] presents a collaborative cloud-edge-end computing framework for improving the efficiency of task processing in mobile-edge computing networks. The proposed framework involves the partial processing of computation-intensive and latency-sensitive tasks at terminals, edge servers, and the cloud. To address the optimization problem of minimizing the sum latency of all mobile devices, a pipeline-based offloading scheme is introduced. The non-convex problem is transformed into a convex one using the SCA (successive convex approximation) approach. Simulation results demonstrate that the collaborative offloading scheme with the pipeline strategy outperforms other offloading schemes in terms of efficiency. Additionally, the article highlights the need for further optimization and updates of conventional approaches such as power allocation, computation offloading/allocation, handover, and subcarrier allocation to support future environment requirements in mobile-edge computing networks.

## 3. Business Framework

Based on the design concepts of separation of concerns, lightweight, and scalability, we proposed a workflow model based on user-level threads and message channels, separating the process control module from the control plane and deploying it as a separate workflow composition engine to the data plane to connect various services. This reduces the time required for starting and recycling each process container and the execution efficiency is equivalent to that of native systems. The framework is shown in Figure 1.

The functional components of the industrial microservice composition business framework include application construction, application store, component repository, etc. The functional architecture is shown in Figure 2.

The service objects of the industrial microservice composition business framework include developers and operational service providers. Different entry points need to be set for different purposes of service objects. We provided a user-friendly and easy-to-use front-end workflow drawing interface. The component repository is the foundation for the enterprise development of microservice applications. Enterprises can retrieve the latest industrial Internet components from the component repository and combine them into the desired application on the workflow drawing interface while also supporting edge device registration and adding to the workflow. After the enterprise completes the workflow-based application construction, it can be published in the customer-facing application store. Customers can download and deploy the corresponding application from the application store.

## 4. Technical Architecture

In response to various problems arising from the combination of Kubernetes clusters in cloud and edge, we proposed a lightweight microservice workflow platform architecture (indicated by the black dotted box in Figure 3). The main components are the workflow drawing front, control plane, and data plane. The implementation of this architecture is shown in Appendix A.

Firstly, in response to the demand for an all-in-one solution for industrial Internet platforms and the existing problems, this architecture solves them by extending the control plane in the form of plugins. By accessing cloud databases, log management modules and monitoring modules, it ensures the data security and stability of the platform. Harbor is used as the image repository to manage private images used in the workflow. Secondly, to solve the problem of complex platform installation and deployment, this architecture uses Helm as the deployment tool, which can deploy all controllers and dependencies with one click. Thirdly, to address the problem of complex workflow application maintenance, Kubernetes native CRD (Custom Resource Definition) is used as the orchestration unit, and the lifecycle of all applications is managed through the mature Kubernetes system, increasing the usability and reliability of application maintenance. Finally, in response to the problem of edge device management and cloud microservice integration, this architecture combines open-source device access frameworks to deploy device data synchronization controllers in the edge, synchronizing device data to cloud CRDs for users to manage devices in a cloud-native way. The Astermule workflow engine is used to unify the data flow between the cloud and the edge, filling the data gap between the cloud and the edge.

The lightweight microservice composition technology architecture we proposed in this article is shown in Figure 3.

The workflow drawing front end is responsible for providing users with a friendly workflow design interface, making it easy for users to draw microservice workflows like flowcharts and providing essential workflow management functions such as deployment, saving, and loading.

The control plane project, named Astertower, integrates components such as backend service gateways, workflow application management, private image management, and workflow engine controllers. The backend service gateway provides essential access control functions for backend applications. The workflow application management integrates cloud databases to store workflow information. The private image management component, which is called Harbor, provides users with a private image repository to develop their own private microservice applications. The workflow engine controller is the core of the entire control plane, responsible for deploying specific workflow application nodes and controlling Astermule (the self-developed workflow engine of this platform), which is used to manage the deployment and execution of workflows.

The data plane consists of multiple services that correspond to multiple workflows and multiple Astermule workflow engines. Each workflow’s services correspond to an Astermule. Astermule exists in the form of containers and is injected with the corresponding workflow information and bound to the workflow by the workflow engine controller in Astertower when it starts. Afterwards, Astermule is responsible for the data flow communication between various microservices and devices during the execution of the workflow.

### 4.1. Business Process Model

Based on BPMN (Business Process Model and Notation), this article proposes a microservice workflow design framework that is based on BPMN 2.0. BPMN is a standard symbols and notation language used for business process modeling. Using BPMN can clarify business processes, facilitate communication and collaboration, improve efficiency, optimize business processes, and support automation. This tool can help enterprises better manage and optimize their business processes, improve efficiency, reduce costs, and increase customer satisfaction.

### 4.2. Astertower Control Plane

In the control layer, we designed and developed the Astertower control plane. The Astertower control plane mainly serves as the backend and Kubernetes controller, and its core is the orchestration of resources required for workflow execution. The Astertower control plane takes on the tasks of workflow parsing and node deployment while also creating an Astermule workflow engine with a corresponding mode that is deployed as a Kubernetes Pod instance.

#### 4.2.1. Building Backend Services and Parsing Workflows

As the backend, Astertower is responsible for converting the workflow description file sent by the front end into a yaml format file. During the parsing process, it checks the correctness of the workflow (whether there are loops in the diagram describing the workflow, whether the workflow nodes have duplicate names, etc.). In addition, it is also responsible for functions such as saving and loading workflows.

Astertower registers a custom resource called Astro with Kubernetes, which corresponds to each workflow. The goal of the backend service for parsing workflows is to create Astro resource files. The definition of an Astro resource file is shown in Figure 4.

#### 4.2.2. Astertower Controller

The core function of the Astertower control plane is to register the custom resource Astro (which is a Kubernetes custom resource that describes workflows and follows the Kubernetes API extension standard) and the custom controller with Kubernetes and manage the orchestration of workflows in a native Kubernetes way. The controller architecture is shown in Figure 5.

Before starting the Astermule workflow engine, the Astertower control plane must ensure that all required microservices and edge devices are ready. The edge devices are synchronized as Kubernetes custom resources by OpenYurt, and the Astermule engine shown Figure 5 accesses the API server to obtain the necessary device information. Therefore, each time the Astertower controller creates an Astro workflow resource, it creates a Deployment instance and a Service corresponding to the workflow node in Kubernetes and adds an Owner Reference to enable Kubernetes to perform garbage collection when the workflow is deleted.

In the design of the execution process of the controller, we propose seven phases of the controller. These seven stages are listed in Table 1.

These seven phases contain all the control logic executed by a workflow. The Astertower controller obtains the current phase of Astro by accessing the phase field of Astro CRD and executes different logic according to different phases, as shown in Figure 6 and Algorithm 1.
**Algorithm 1.** Astertower controller reconciling process**Input:** The key to getting the astro needed reconciling**Output:** A reconciling completed *Initialization:* *Register Astro CRD with Kubernetes.**Register with kube-apiserver to monitor Astro resources, Deployment resources, and Service resources.* *Launch the Astertower controller as a Pod in Kubernetes. The controller applies for multiple Go threads as workers.* *Each change of watched resources causes the Astro to enter a rate-limiting queue, waiting to be picked up and executed by an idle worker.* *The following code is the logic for each worker’s execution.*A change event occurred and Astro that needs to be synchronized was obtainednamespace, name:=cache.SplitMetaNamespaceKey(key)*Get an Astro object (workflow object) information from the waiting queue using key*3.astro, err:=c.astroLister.Astros(namespace).Get(name) *Use this information to get the real object from the kube-apiserver*4.**if** (astro.DeletionTimestamp.IsNotZero()) **then**5.delete(astro)6.**end if***Each deleted Astro object is time-stamped and scheduled to the worker to perform the actual deletion action*7.**else if** (astro.HasFinalizer(AstroFinalizer)) **then**8.**switch** astro.status.phase9.**case** AstroPhaseInitialized:10.**if** all status of resources is Ready **then**11.astro.status.phase = AstroPhaseWaited12.**else**13.astro.status.phase = AstroPhaseDeployFailed14.**end if***In the initialization phase, it is necessary to check whether the deployment and service corresponding to all nodes are completed. If they are completed, they can enter the Wait phase*15.**case** AstroPhaseWaited:16.**if** (astro.status.AstermuleRef.name.isNil()) **then**17.astro.status.astermuleRef = newAstermule(astro.namespace,astro.name)18.**else**19.pod = GetPod (astermuleRef.name)20.**if** pod.status.phase = corev1.PodRunning **then**21.astro.status.phase = AstroPhaseReady22.**else if** pod.status.phase = corev1.PodFailed **then**23.astro.status.phase = AstroPhaseEngineFailed24.**end if**25.**end if***The waiting phase starts the workflow engine as a pod*26.**case** AstroPhaseDeployFailed:27.HandleDeployError()*Deploy error occurred, error handling*28.**case** AstroPhaseEngineFailed:29.HandleEngineError()*Engine error occurred, error handling*30.**case** AstroPhaseReady:31.astermule = astro.status.astermuleRef32.pod = GetPod(astermuleRef.name)33.url = makeURL(pod.status.PodIP)34.astro.status.result = sendCommand(url)35.**if** astro.result.health = true **then**36.astro.status.phase = AstroPhaseSuccess37.**else**38.astro.status.phase = AstroPhaseWrong39.**end if***Entering the Ready phase means that all the dependent resources and the workflow engine have been deployed. We need to find and send a request to the startup URL of the astermule workflow engine, which will then return the result to us*40.**else**41.create(astro)42.**for each** deployment in astro.spec **do**43.if (NotExists(deployment)) then44.create(deployment)45.**end if**46.**end for**47.**for each** service in astro.spec **do**48.**if** (NotExists(service)) **then**49.create(service)50.**end if**51.**end for**52.astro.status.phase = AstroPhaseInitialized53.AddFinalizer(astro)*The first Astro creation goes into this branch and sends the deployment commands for all the required resources to Kubernetes*54.**end if**

### 4.3. Astermule Workflow Engine

At the data plane, we designed and developed Astermule workflow engines, each of which exists in a Pod instance form for each specific workflow. After receiving a request from the control plane, the Astermule engine accesses the corresponding Service for each Deployment concurrently according to the process and returns the final execution result to the control layer.

#### 4.3.1. Lightweight and High-Concurrency Design

Based on the lightweight design philosophy, the Astermule workflow engine aims to minimize additional overhead during workflow execution. In traditional CICD workflows, a separate Pod is started for each node, which results in significant performance overhead in the context of service composition. Therefore, Astermule uses a single-process model to perform service composition, starting only one goroutine for each node (A–D in Figure 7). The goroutine accesses the corresponding microservice data and device data, and data is passed between goroutines using Go channels, which are used to pass data in the Go language’s concurrency mechanism. The design of the Astermule workflow engine is shown in Figure 7.

To handle errors that may occur when each goroutine accesses its corresponding microservice as a client, Astermule uses an error propagation mechanism. The data passed to each goroutine after the data is processed indicates whether the current node’s access was successful. If the following goroutine finds out that the previous goroutine’s access was not successful, it will propagate this information to its successor goroutine. This way, when the workflow is completed, it will be known whether the execution has failed.

#### 4.3.2. Astermule Execution Process

The data plane consists of many Pod instances of the Astermule components, each of which corresponds to a workflow and is the core component for data transmission during workflow execution. It is also a key component for lightweight microservice composition in this platform. The design architecture of the Astermule component does not depend on Kubernetes or any specific environment. It is an independent component whose behavior pattern involves receiving a json file containing node relationships and the address of the corresponding node service as command-line parameters. After reading the file, it switches to server mode and waits for an instruction. When it receives the corresponding instruction, it starts accessing each node in order according to the topology and ensures that the data from the previous node can be accurately sent to the next node. The Astermule engine execution logic is shown in Figure 8 and Algorithm 2.
**Algorithm 2.** Astermule workflow engine execution process**Input:** Node information, deployment and service information for each cloud node, and metadata information of each device node**Output:** Workflow execution result *Initialization:* *After startup, wait until the controller sends a request to start executing the following logic.*1.type Message struct {2.Status Status ‘json:”status”’3.Data string ‘json:”data”’4.}*Defines a Message structure for passing state and data between nodes*5.**for each** node in nodes **do**6.**for each** dep in node.dependencies **do**7.ch:=make(chan Message)8.channelGroup[dep.name].WriteChannel = append(channelGroup[dep.name].WriteChannel,ch)9.channelGroup[node.name].ReadChannel = append(channelGroup[dep.name].ReadChannel,ch)10.**end for**11.**end for***Build a directed edge between nodes, putting read-only channels and writing-only channels into both ends of the directed edge*12.functionSet:=make([]func())13.**for each** node in nodes **do**14.chGrp:=channelGroup[node.name]15.function:=func() {16.msgs:=make([]Message)17.**for each** readChannel in chGrp.ReadChannel **do**18.msg:=<-readChannel19.msgs = append(msgs,msg)20.**end for***A node must read data from all read-only channels to ensure that all of its predecessor nodes are executed*21.mergeMsg:=&Message{}22.**for each** msg in msgs **do**23.msg.DeepMergeInto(mergeMsg)24.**end for***Data from multiple precursor nodes needs to be merged*25.sendMsg:=&Message{}26.sendMsg.status.health = true27.res,err:=Send(node.action,node.url,mergeMsg.data)*Send a request to the url of the current node and send the combined data as input to obtain the data of the current node. In the specific code, the Send function needs to distinguish whether the local node is a device node. If it is a device node, it needs to obtain the information from the corresponding device address*28.**if** (err! = nil) **then**29.sendMsg.status.health = false30.**else**31.sendMsg.data = res32.**end if***An error propagation mechanism that propagates an error down a node if an error occurs*33.**for each** writeChannel in chGrp.WriteChannel **do**34.writeChannel<-*sendMsg35.**end for***Sends message to all successor nodes*36.}37.functionSet = append(functionSet, function)*Each node corresponds to a function object, and the entire graph is constructed as a set of function objects*38.**end for**39.execution(functionSet) *Assign each function to a coroutine and start it*40.sendMsgToInitNode()*Sending messages to the channel of nodes that have no precursor launches the entire graph*

After creating all the Deployments and Services mentioned above, the Astertower controller uses Kubernetes’ list-watch function to monitor these resources, ensuring that all corresponding Service ClusterIPs (used by Kubernetes to load-balance internal services, where the IP of the Service is the entry point to the corresponding service) are collected only when all resources are ready. These Service IPs, together with the port and target of each node described in the Astro resource in Figure 5, are combined into URLs for each node service. These URLs are then deployed as startup parameters for Astermule so that it can access the entry points of all service nodes.

For device nodes, the process of obtaining URLs is slightly different. Device nodes exist in the OpenYurt cluster as custom resources called “Device,” so the corresponding path of the Device resource in Kubernetes only needs to be specified when filling in the URL. With RBAC (Kubernetes’ role-based access control) configured, Astermule can access the kube-apiserver (Kubernetes’ API server) to retrieve device data.

#### 4.3.3. Workflow Engine Model

In the traditional CICD workflow, there are mainly three stages. Firstly, the container of the node to be executed is started, assuming that the time consumed by this operation is $T_{s}$. After the container is started, the specific logic of the node is executed, assuming that the time consumed by this stage is $T_{p}$. Finally, the node needs to be destroyed and the time occupied by resource recycling is $T_{e}$. Therefore, the time for executing n nodes in the CICD workflow is:(1)$$n\times \left(T_{s}+T_{p}+T_{e}\right)$$

In the workflow execution model of the Astermule engine, since the startup and destruction of workflow nodes are uniformly managed by the Operator, the startup and destruction of all nodes can be concurrent. In the ideal state, the time for n nodes to start and destroy containers only needs $T_{s}+T_{e}$. For the processing of specific node logic, Astermule uses Golang coroutines and channels for control. Assuming that the time to create and destroy these coroutines is $T_{g}$ and $T_{c}$, the time for Astermule workflow engine to execute n nodes is:(2)$$T_{s}+T_{e}+n*T_{p}+T_{g}+T_{c}$$

In summary, the optimization of Astermule engine for workflow execution is:(3)$$n\times \left(T_{s}+T_{e}-T_{g}-T_{c}\right)$$

It is known that at the operating system level, the operations performed to create and destroy containers are much greater than those performed to create and destroy coroutines. Therefore, $T_{s}+T_{e}>>T_{g}+T_{c}$. The optimization of Astermule for execution time is theoretically very significant, and it will achieve a tremendous improvement in the case of a large number of nodes.

## 5. Results

To test the effectiveness of the Astertower platform in achieving lightweight and efficient workflow execution, this article designed the following three test scenarios for four-node, six-node, and ten-node workflows.

This test uses an Alibaba Cloud server with a 4-core CPU and 4GB of memory, running the Ubuntu 22.04 operating system. The Kubernetes cluster used in the test was simulated using the Kind tool, which included one master node and two worker nodes. In the nine scenarios of three different node counts and three different workflow concurrency levels, the platform was used to execute the workflows. For this experiment, test nodes were written for each node, and each node started a service on http://localhost:8000/test, (accessed on 30 May 2023). that accepted any form of JSON. The service increased the JSON’s hostname as the name field. The configurations required for this experiment are shown in Table 2.

We chose Argo-Workflow as our baseline. Argo Workflows is a well-known open-source container-native workflow engine for orchestrating parallel jobs on Kubernetes. The results were compared by running each workflow ten times and taking the average. Since the workflows were executed on a single-node simulated cluster, the execution time was relatively long, so the time data was rounded off, as shown in Table 3 and Figure 9.

According to the data in the table, it can be seen that compared to the baseline of Argo-Workflow that requires starting a Pod for each node, the workflows executed by Astertower and Astermule had much shorter execution times, and this advantage becomes more pronounced as the number of nodes increases.

## 6. Conclusions

This article is mainly based on the concept of lightweight microservice composition to explore the architecture design of a lightweight microservice composition platform for cloud-edge integration. With the ability to access devices, a lightweight workload engine and corresponding controller were developed for cloud-edge scenarios. Using the extensibility of Kubernetes custom resources, the workflow execution process is orchestrated and monitored in Kubernetes. A front-end visual microservice workflow design interface was implemented to make the platform easy to use and manage while fully integrating Internet resources and industrial manufacturing resources.

Compared to traditional industrial internet integration, which faces difficulties in resource integration, complex interfaces, and difficulty in accessing edge devices, this article presents a design for a control plane and data plane (workflow engine) specifically tailored for cloud-edge scenarios. And by stripping heavy container runtimes at the edge, tasks are executed on highly concurrent user-level threads instead, resulting in a lightweight and efficient architecture that achieves a significant time advantage over traditional CICD workflows. This design can bridge the gap between cloud and edge workflows and achieve a more efficient and unified orchestration system.

## Figures and Tables

**Figure 1 sensors-23-05939-f001:**
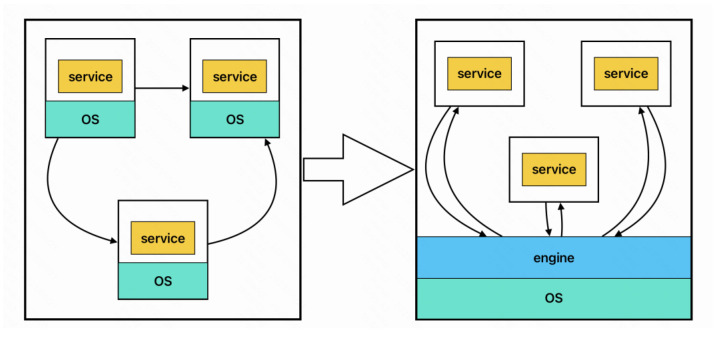
Lightweight workflow composition model.

**Figure 2 sensors-23-05939-f002:**
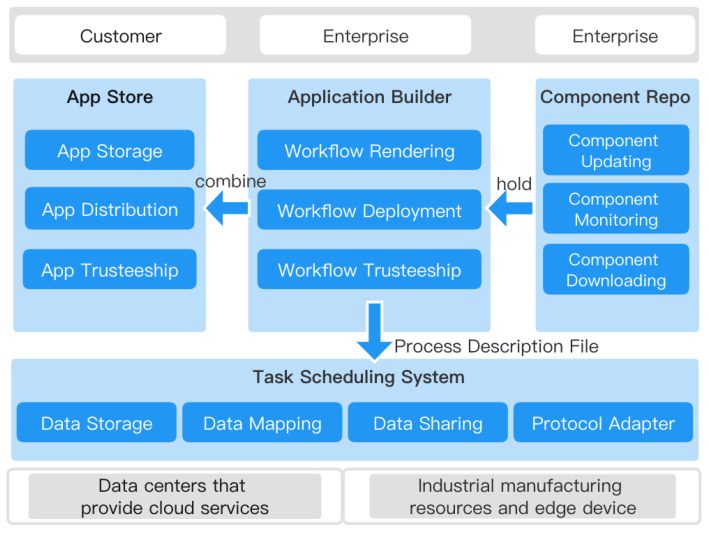
Industrial microservices portfolio business framework.

**Figure 3 sensors-23-05939-f003:**
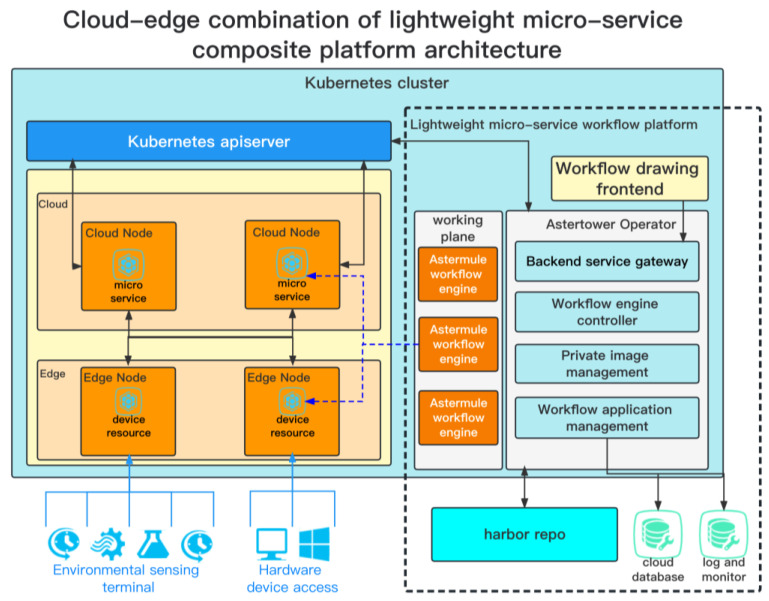
Cloud-edge combination of lightweight micro-service composite platform architecture.

**Figure 4 sensors-23-05939-f004:**
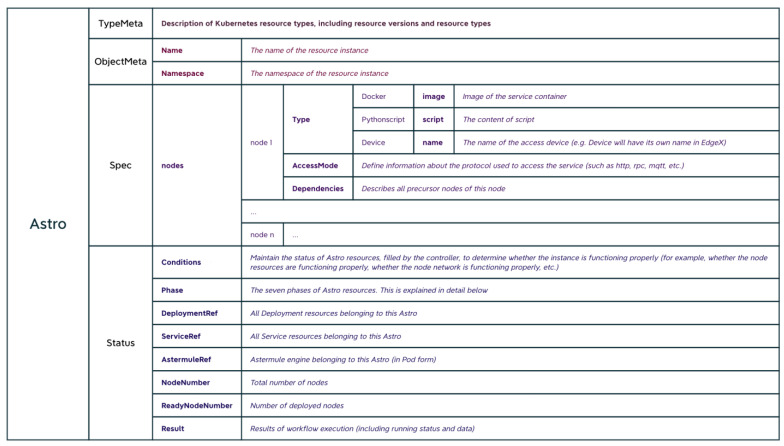
Definition of Astro custom resources.

**Figure 5 sensors-23-05939-f005:**
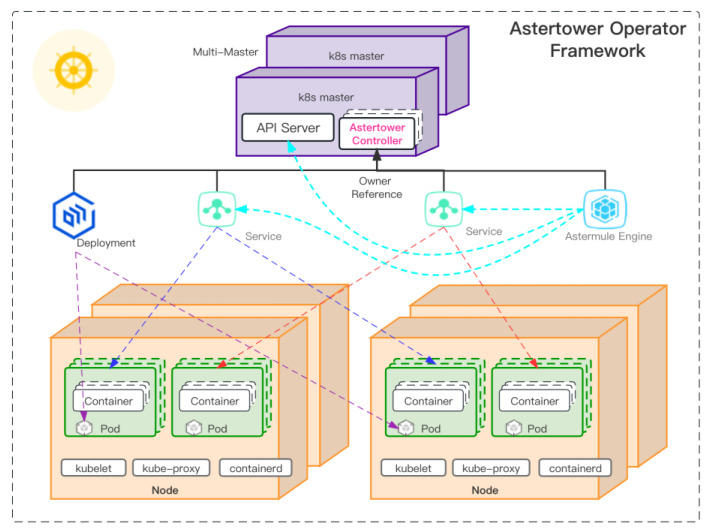
Astertower controller.

**Figure 6 sensors-23-05939-f006:**
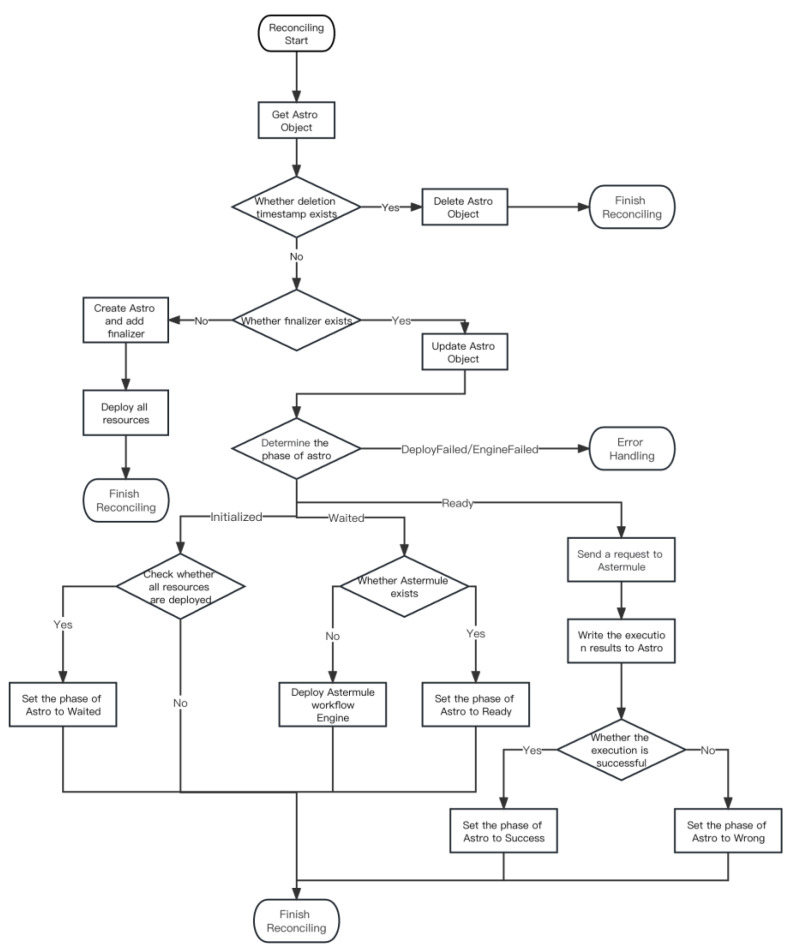
Astertower controller reconciling flowchart.

**Figure 7 sensors-23-05939-f007:**
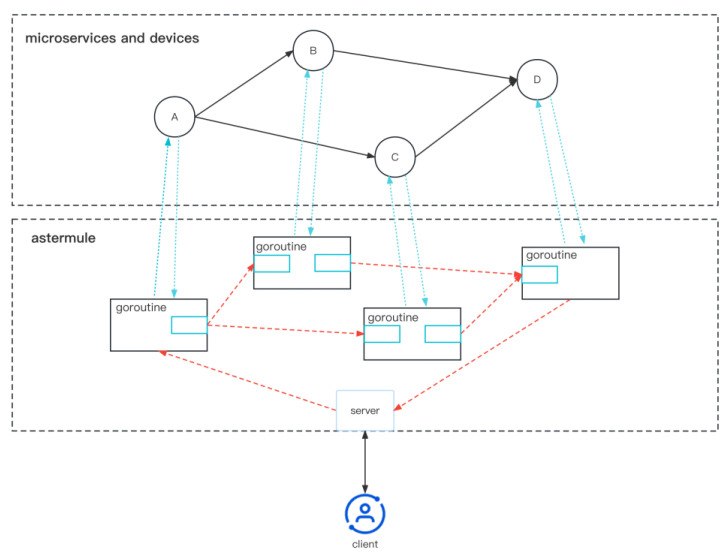
Astermule Lightweight workflow engine architecture.

**Figure 8 sensors-23-05939-f008:**
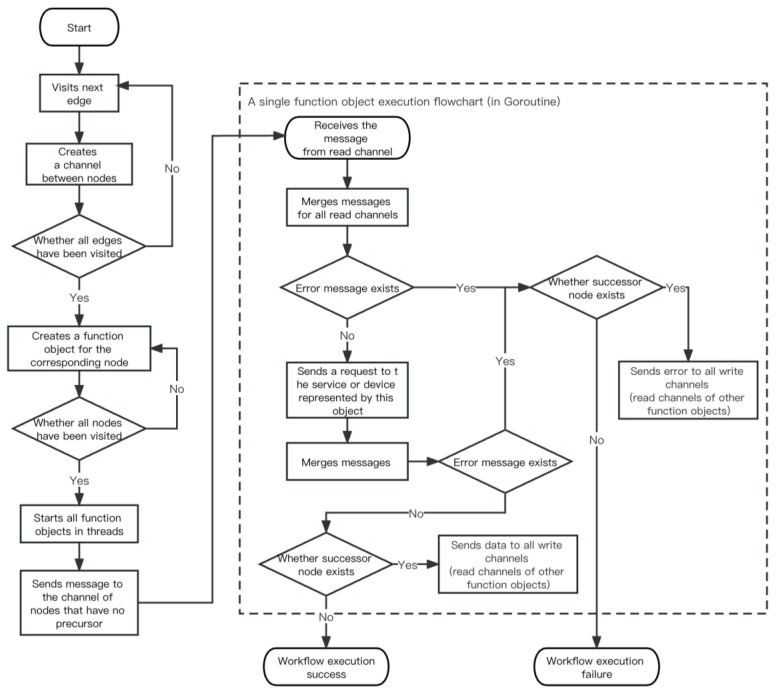
Astermule workflow engine execution flowchart.

**Figure 9 sensors-23-05939-f009:**
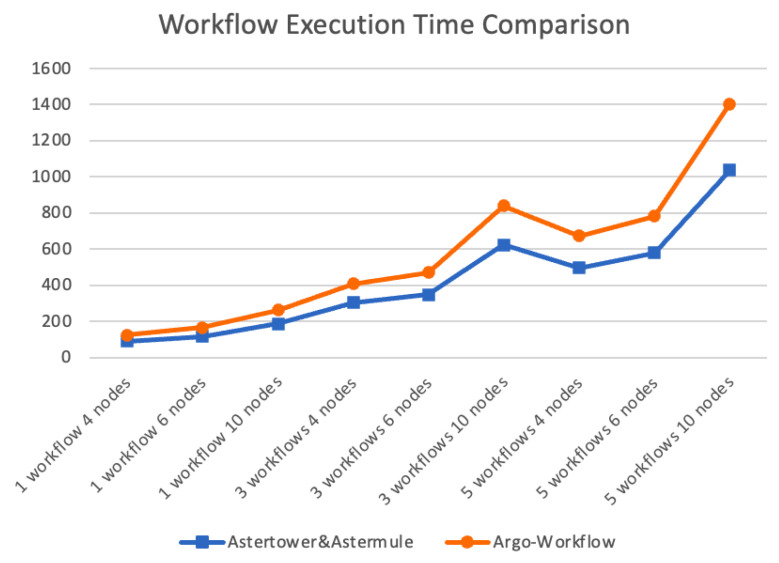
The efficiency line chart of the two workflow engines.

**Table 1 sensors-23-05939-t001:** The phase definition of Astro CRD.

Phase	Definition
Initialized	Astro CRD Reconciling for the first time
Waited	Wait for the required deployment and service to be deployed
DeployFailed	The required deployment and service deployment failed
EngineFailed	Astermule engine deployment failed
Ready	The required resources and engines are deployed successfully. Wait for the controller to send a request to the engine
Success	The workflow engine Astermule executed successfully and the result was written to the result field of the Astro CRD
Wrong	Astermule workflow engine execution failure

**Table 2 sensors-23-05939-t002:** Test environment setting list.

Environment	Settings
Cloud Platform	Alibaba Cloud Server
CPU	4-core (sharing type)
Memory	4 GB
Operating System	Ubuntu 22.04
Kind	v0.17.0
Kubernetes	v1.22.15
Node Image	kasterism/test_a:latest (dockerhub)
Astertower Version	v0.1.0
Astermule Version	v0.1.0

**Table 3 sensors-23-05939-t003:** Workflow execution time comparison table.

Workflow Platform	Number of Workflow Concurrent	Four Nodes	Six Nodes	Ten Nodes
Argo-Workflow	1	89.054s	116.756s	185.976s
	3	304.430s	347.973s	623.936s
	5	495.875s	578.931s	1037.808s
Astertower and Astermule	1	33.294s	47.472s	75.689s
	3	103.485s	122.734s	214.324s
	5	175.878s	201.846s	366.149s

## Data Availability

The experiment in this article does not require specific data. Please refer to Appendix A for code and environment.

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
