# Peer review of "Research on Lightweight Microservice Composition Technology in Cloud-Edge Device Scenarios"

_sensors, 2023, doi:10.3390/s23135939_

Round 1
Reviewer 1 Report
In this paper, the authors present Research on lightweight microservice composition technology in cloud-edge device scenarios. The paper presents a lightweight micro-service composite platform architecture that supports the combination of cloud and edge.
1. The language needs minor review to improve readability.
2. The Abstract section should summarize in a very concise way the main achievement of the paper. Describe vague ideas about the main results. The authors are instigated to highlight what is new in their studies.
3. 0. Introduction????
4. The novelty of this paper is not clear. The difference between present work and previous Works should be highlighted.
5. Authors must develop the framework/architecture of the proposed methods
6. There is need of flowchart and pseudocode of the proposed techniques
7. Proposed methods should be compared with the state-of-the-art existing techniques
8. In case to be allowed, it would be interesting authors create a list of abbreviatures.
9.Though I could understand almost everything from every line as authors would have intended, however I would request authors to once again re-look into the grammer and unnecessary typos.
10. Some of the definitions included in the article should be improved for a better understanding by the reader since some are very basic.
11. Minor: keep the format in order to cite figures: Figure X and not figure x.
Finally, paper needs major improvements
In this paper, the authors present Research on lightweight microservice composition technology in cloud-edge device scenarios. The paper presents a lightweight micro-service composite platform architecture that supports the combination of cloud and edge.
1. The language needs minor review to improve readability.
2. The Abstract section should summarize in a very concise way the main achievement of the paper. Describe vague ideas about the main results. The authors are instigated to highlight what is new in their studies.
3. 0. Introduction????
4. The novelty of this paper is not clear. The difference between present work and previous Works should be highlighted.
5. Authors must develop the framework/architecture of the proposed methods
6. There is need of flowchart and pseudocode of the proposed techniques
7. Proposed methods should be compared with the state-of-the-art existing techniques
8. In case to be allowed, it would be interesting authors create a list of abbreviatures.
9.Though I could understand almost everything from every line as authors would have intended, however I would request authors to once again re-look into the grammer and unnecessary typos.
10. Some of the definitions included in the article should be improved for a better understanding by the reader since some are very basic.
11. Minor: keep the format in order to cite figures: Figure X and not figure x.
Finally, paper needs major improvements
Author Response
Thanks very much for your time to review this manuscript. I really appreciate all your comments and suggestions. We have considered these comments carefully and tried our best to address every one of them.
Response to Reviewer 1:
Comment 1: The language needs minor review to improve readability.
Response 1: Thank you for your suggestion. We have made comprehensive changes to the readability of the entire article to make the language more refined and accurate. Please see the attachment.
Comment 2: The Abstract section should summarize in a very concise way the main achievement of the paper. Describe vague ideas about the main results. The authors are instigated to highlight what is new in their studies.
Response 2: Thank you for your suggestion. We have rewritten the second half of the abstract to highlight the innovation and novelty of the research. Please see the attachment.
Comment 3: 0. Introduction????
Response 3: We are very sorry that we were not very professional in the layout of the title before, and now all the numbers before the title have been re-arranged. Please see the attachment.
Comment 4: The novelty of this paper is not clear. The difference between present work and previous Works should be highlighted.
Response 4: Thank you for your suggestion. We add a paragraph at the end of Introduction to explain the difference between the work in this paper and the previous work, and summarize the innovation points again in the Conclusion paragraph. Please see the attachment.
Comment 5: Authors must develop the framework/architecture of the proposed methods.
Response 5: Thank you for your suggestion. We have initially developed a platform, including operator and workflow engine. You can find links to them in the new Appendix B. Please see the attachment.
Comment 6: There is need of flowchart and pseudocode of the proposed techniques.
Response 6: Thank you for your suggestion. We have added corresponding flowchart and pseudocode in both 4.2.2 and 4.3.2 (includes the logic of the controller and the logic of the workflow engine), which can really help others understand our design idea better. Please see the attachment. Thank you again for your suggestion.
Comment 7: Proposed methods should be compared with the state-of-the-art existing techniques.
Response 7: Thank you for your suggestion. We compared our workflow with the well-known Argo-Workflow in Results, added the introduction of Argo-Workflow in this revision, explained the environment of the experiment in more detail, and redrew the line chart. Please see the attachment.
Comment 8: In case to be allowed, it would be interesting authors create a list of abbreviatures.
Response 8: Thank you for your suggestion. We added A list of abbreviations to Appendix A. Please see the attachment.
Comment 9: Though I could understand almost everything from every line as authors would have intended, however I would request authors to once again re-look into the grammer and unnecessary typos.
Response 9: Thank you for your suggestion. We fixed a lot of syntax errors and typos in this revision. Please see the attachment.
Comment 10: Some of the definitions included in the article should be improved for a better understanding by the reader since some are very basic.
Response 10: Thank you for your suggestion. In this revision, we revisited our definition of some concepts and processes. In 4.2.1 we added a detailed description of the Astro resource definition (in Figure 4), and in 4.2.2 we added a description of the various states of Astro during synchronization. Hopefully this will improve some of the definition issues. Please see the attachment.
Comment 11: Minor: keep the format in order to cite figures: Figure X and not figure x.
Response 11: Thank you for your suggestion. We checked all references in this revision, and now they use Figure uniformly. Please see the attachment.

Reviewer 2 Report
This paper proposes a lightweight micro-service composite platform architecture for small and medium-sized enterprises.
1. The authors should add their main evaluation performance in the abstract.
2. The authors should summarize their main contributions and novelty in the introduction section.
3. The related work discussion is not comprehensive. The authors should add more discussion for recent relevant publications, e.g., those published in the last three years.
4. It is hard to tell which parts are proposed by the authors and which parts are borrowed from off-the-shelf software/framework, e.g., Alibaba Cloud and Kubernetes.
5. Figure 8 has a low resolution and should be improved.
6. The authors should add the comparison with existing solutions or similar studies as baselines in the numerical experiments.
Author Response
Thanks very much for your time to review this manuscript. I really appreciate all your comments and suggestions. We have considered these comments carefully and tried our best to address every one of them.
Response to Reviewer 2:
Comment 1: The authors should add their main evaluation performance in the abstract.
Response 1: Thank you for your suggestion. We have rewritten the second half of the abstract to highlight the innovation and novelty of the research, and have increased our execution time and resource utilization advantages over traditional workflows. Please see the attachment.
Comment 2: The authors should summarize their main contributions and novelty in the introduction section.
Response 2: Thank you for your suggestion We added a paragraph at the end of Introduction to explain the difference between the work in this paper and the previous work and highlighted the innovation and novelty of our research. Please see the attachment.
Comment 3: The related work discussion is not comprehensive. The authors should add more discussion for recent relevant publications, e.g., those published in the last three years.
Response 3: Thank you for your suggestion. We have added a description of various optimization studies for cloud-edge scenarios in 2.4, which is based on two IEEE papers and is relatively new. Please see the attachment.
Comment 4: It is hard to tell which parts are proposed by the authors and which parts are borrowed from off-the-shelf software/framework, e.g., Alibaba Cloud and Kubernetes.
Response 4: Thank you for your suggestion. we made some changes to the description of our work, using "we..." when describing our proposed design in 4.2 and 4.3. And we added detailed flowcharts and pseudocode in 4.2.2 and 4.3.2. Please see the attachment.
Comment 5: Figure 8 has a low resolution and should be improved.
Response 5: Thank you for your suggestion. We have redrawn Figure 8 and now it is clear. Please see the attachment.
Comment 6: The authors should add the comparison with existing solutions or similar studies as baselines in the numerical experiments.
Response 6: Thank you for your suggestion. We compared our workflow with the well-known Argo-Workflow in Results, added the introduction of Argo-Workflow in this revision, explained the environment of the experiment in more detail, and redrew the line chart. We also added more experimental environment information and code repository (Appendix B). Please see the attachment.

Reviewer 3 Report
This paper addresses the challenges faced by small and medium-sized enterprises in the Internet industrial manufacturing sector during digital transformation, including integration difficulties and slow development processes. The proposed solution is a lightweight platform that combines cloud and edge technologies, integrating control, work, and workflow components. It utilizes open-source edge computing frameworks for data orchestration and enables the integration of various functions. The platform employs Kubernetes and a self-developed lightweight workflow engine for efficient microservice composition and development on the cloud side.
This paper is well-organized and described the proposed idea well; however, there are several concerns to address including:
1. In Fig. 4, authors should present the created and detail customized Astro resource files for this particular problem (instead of an example), since it is mainly placed in Section 3. And Fig. 6 presents only the key and values of the testing format; which I believe that it is not comprehensive enough by just expressing the getting service-ip and port from A and posting back to B. Please kindly improve the figure purposes (e.g., extending the key/value complexity or various hyper/parameters))
2. Before expressing the results, authors are suggested to add the API setup details (e.g., runtime, integration, adapter, etc.) The interactions between edge and cloud entities should be further expressed in terms of API configuration.
3. Are there any overhead latency in this proposed system? and is it auto-scaling?
4. (optionally) authors are highly recommended to cite the following papers that mention how the working flow of collaborative cloud and edge virtualization and related terms on resource placement:
- https://doi.org/10.1109/TCCN.2020.3018159
- https://doi.org/10.1109/TNSE.2022.3200057
Author Response
Thanks very much for your time to review this manuscript. I really appreciate all your comments and suggestions. We have considered these comments carefully and tried our best to address every one of them.
Response to Reviewer 3:
Comment 1: In Fig. 4, authors should present the created and detail customized Astro resource files for this particular problem (instead of an example), since it is mainly placed in Section 3. And Fig. 6 presents only the key and values of the testing format; which I believe that it is not comprehensive enough by just expressing the getting service-ip and port from A and posting back to B. Please kindly improve the figure purposes (e.g., extending the key/value complexity or various hyper/parameters))
Response 1: Thank you for your suggestion. We have improved Figure 4 and now it is not a simple example but a complete table describing the fields of the Astro resource. Moreover, we have added the various state definitions for the Astro CRD and their transformation logic (flowchart and pseudocode in 4.2.2), which helps the reader to understand the details of the Astro resource more clearly. Please see the attachment.
Comment 2: Before expressing the results, authors are suggested to add the API setup details (e.g., runtime, integration, adapter, etc.) The interactions between edge and cloud entities should be further expressed in terms of API configuration.
Response 2: Thank you for your suggestion. We added a section to the Results that describes the input and output of the node and the API access addresses of the node, and listed the images used by the node in the experimental environment table. Please see the attachment.
Comment 3: Are there any overhead latency in this proposed system? and is it auto-scaling?
Response 3: Thank you for your suggestion. The main latency in this system is network (network communication between hosts), but network latency also exists in CICD workflows (often when hosts need to access object storage), and distributed systems inevitably encounter this problem. Because of the reduced overhead of starting and destroying containers, the workflow can be run with less latency than traditional workflows. In terms of automatic scaling, this system mainly relies on the native capabilities of Kubernetes. For example, when deploying cloud service resources, we use Deployment+Service workload, so the number of instances can be set by adjusting replicas parameters. The Astermule engine is currently in Pod form so there's no way to scale automatically, but it's not hard to do.
Comment 4: (optionally) authors are highly recommended to cite the following papers that mention how the working flow of collaborative cloud and edge virtualization and related terms on resource placement.
Response 4: Thank you for your suggestion. We have added references to these two articles and an introduction to the work of these two articles in 2.4. Please see the attachment.

Round 2
Reviewer 2 Report
Dear authors,
Thanks for revising and resubmitting the manuscript. Previous problems are solved and no further questions.